# Mesenchymal Stem Cells Modulate Granulosa Cell Function Under Inflammatory and Hypoxic Conditions

**DOI:** 10.3390/biomedicines14010027

**Published:** 2025-12-22

**Authors:** Kalina Belemezova, Milena Kostadinova, Tsvetelina Oreshkova, Ivaylo Vangelov, Maria Yunakova, Tanya Timeva, Ivan Bochev

**Affiliations:** 1Department of Biology, Medical Faculty, Medical University—Sofia, 1431 Sofia, Bulgaria; 2Department of Molecular Immunology, Institute of Biology and Immunology of Reproduction, Bulgarian Academy of Sciences, 1113 Sofia, Bulgaria; milena_kostadinova@yahoo.com (M.K.); tsveti_oreshkova@yahoo.com (T.O.); lakatush@yahoo.com (I.B.); 3Department of Immunobiology of Reproduction, Institute of Biology and Immunology of Reproduction, Bulgarian Academy of Sciences, 1113 Sofia, Bulgaria; v_angel_off@abv.bg; 4IVF Department, Ob/Gyn Hospital Dr. Shterev, 1330 Sofia, Bulgaria; m_yunakova@yahoo.com (M.Y.); ttimeva@yahoo.com (T.T.); 5Department of Obstetrics and Gynecology, Medical University—Sofia, 1431 Sofia, Bulgaria; 6Department of Health Care, University of Ruse, 7017 Ruse, Bulgaria

**Keywords:** mesenchymal stem cells, granulosa cells, premature ovarian insufficiency, inflammation, hypoxia

## Abstract

**Background/Objectives:** Increasing evidence points to hypoxia and inflammation as two major causes of compromised ovarian function. Increased oxidative stress under hypoxic conditions can damage cellular components, leading to the dysfunction and apoptosis of granulosa cells (GCs). The inflammatory response induced by hypoxia may further impair the function of the ovaries and contribute to the development of premature ovarian insufficiency (POI). In animal models of premature ovarian failure, research has demonstrated that the transplantation of mesenchymal stem cells (MSCs) can enhance reproductive outcomes, increase the number of functioning ovarian follicles, and restore estradiol production. However, the specific mechanisms underlying the observed positive results are not well understood. **Methods**: The present study provides a comparative analysis of how MSCs influence human GC function under inflammatory and hypoxic conditions, using three different experimental approaches: direct co-culture, indirect co-culture with transwell cell culture inserts, and treatment with MSC-derived conditioned medium (MSCcm). **Results**: Inflammation significantly suppressed GC estradiol secretion and increased apoptosis. MSCs increased estradiol secretion in normal and hypoxic culture conditions when co-cultured directly with GCs. Our results also showed that, under inflammation, MSCs tended to decrease GC proliferation and that hypoxia alone did not have an effect on GC estradiol secretion or proliferation. **Conclusions**: The study emphasizes the dual nature of MSCs, which largely determines their effects on other cell types, and the need for the condition-specific optimization of MSC therapies for ovarian regeneration.

## 1. Introduction

Premature ovarian insufficiency (POI), also known as premature ovarian failure (POF), is a clinical condition characterized by the loss of normal ovarian function before the age of 40. The condition is characterized by amenorrhea, anovulation, hypoestrogenemia, elevated serum gonadotropin levels, a reduced number of ovarian follicles, and infertility. In terms of etiology, primary POI can be both congenital (certain genetic factors, such as Turner syndrome, a premutation for fragile X syndrome, and mutations in genes involved in ovarian function (*FOXL2* and *BMP15*)) and caused by iatrogenic factors such as chemotherapy/radiotherapy or surgical interventions [1]. The risks following certain surgical procedures arise from the likelihood of affecting healthy ovarian tissue, disrupting the blood supply to the ovary, and a subsequent inflammatory reaction. Causes can also be infectious or other diseases, such as endometriosis, including autoimmune diseases [2]. However, POI is most often classified as idiopathic, i.e., without a clear cause. Irrespective of the etiology and causative agent, ultimately, POI is a direct result of two alternative conditions—follicular depletion and follicular dysfunction. Both lead to the generalized dysfunction of the ovaries, where granulosa cells play a critical role in follicular development and hormone production [3]. Many of these causes are also linked to the development of a local inflammatory reaction or impaired blood flow, which in turn leads to a lack of nutrients and low oxygen levels in the ovaries. The ovaries are particularly vulnerable to damage, and any exposure to adverse conditions can be associated with impaired folliculogenesis, chronic disease, and infertility [4].

Granulosa cells (GCs) play a central role in follicle development. Their main function is to produce the key steroid hormones estradiol and progesterone, the concentrations of which vary depending on the stage of the menstrual cycle. During the growth of antral follicles, GC proliferation and differentiation are particularly important in developing sensibility to luteinizing hormone (LH) and communicating with the oocyte. During follicular growth and development, GCs and theca cells undergo specific molecular and functional changes that determine the fates of antral follicles. Another important function of GCs during ovarian folliculogenesis is the control of follicular degeneration, as GC apoptosis is a major mechanism for follicular atresia [5].

One of the most promising approaches for the treatment of POI is autologous therapy with mesenchymal stem cells [6]. The concept of this innovative approach is based on the impressive progress and results achieved in MSC therapy for a number of diseases. These are multipotent stem cells that can be obtained from a variety of tissue sources, in accordance with their wide, probably ubiquitous, distribution in postnatal organs [7]. MSCs have the ability to differentiate into a variety of mesodermal cell types and, under specific inductive conditions, such as exposure to defined growth factors or specialized culture environments, into non-mesodermal lineages [8,9]. Furthermore, their pronounced proliferative potential and anti-inflammatory, anti-apoptotic, and immunomodulatory properties make them an attractive tool for the treatment of degenerative and immune-related diseases [10]. Several clinical reports have been published documenting the use of autologous or allogeneic MSCs for the treatment of POI in women, which resulted in the restoration of ovarian function, increases in anti-Müllerian hormone (AMH), and improved fertility outcomes [11].

With the growing interest in cell-free regenerative therapies, the secretome from MSCs has emerged as a promising substitute for direct cell transplantation that is low-risk and easy to obtain at lower financial costs [12]. All soluble substances that stem cells produce and use for intercellular communication are collectively referred to as the “secretome”. These include serum proteins, growth factors, angiogenic factors, hormones, cytokines, chemokines, and even small amounts of genetic material and lipid mediators [13]. While their therapeutic potential is increasingly acknowledged, the ability of MSCs and their secretome to influence the behavior of other adult cell types, particularly granulosa cells, which are key players in ovarian health and folliculogenesis, are still poorly understood.

## 2. Materials and Methods

### 2.1. Overall Experimental Design

A summary diagram of the experimental design is represented in Figure 1. Mesenchymal stem cells were isolated from bone marrow aspirates and further characterized according to the defined criteria for MSCs—a colony-forming assay, flow cytometry analysis for specific surface antigens, and osteogenic and adipogenic differentiation. Human granulosa cells were isolated from preovulatory follicles and characterized as detailed in the Section 2. Two different culture conditions were set—hypoxic and inflammatory—and three different experimental designs were used: direct co-culture of MSCs and GCs, indirect co-culture of MSCs and GCs using a transwell system, and GCs cultured in the presence of an MSC-derived conditioned medium. The main GC functional characteristics were then examined—their proliferation potential, apoptotic rate, and estradiol secretion.

### 2.2. Follicular Fluid Collection

The preovulatory follicles of 5 women undergoing ovarian follicle puncture as part of an in vitro fertilization (IVF) treatment at the General Hospital Dr. Shterev, Sofia were used to collect follicular fluid (FF) and isolate human primary granulosa cells. Each participant signed a written consent form approved by the Ethics Committee of the General Hospital Dr. Shterev (Sofia, Bulgaria), enabling the collection and use of granulosa cells and follicular fluid for research. Following the removal of cumulus–oocyte complexes from the follicular aspirates, the granulosa cell-containing leftover follicular fluid was moved to sterile 15 mL conical polypropylene tubes (EuroClone, Pero, Italy) and processed within an hour. Samples with high levels of erythrocyte contamination were not included in the study.

### 2.3. Isolation and Cultivation of Human Granulosa Cells

After centrifuging the follicular fluid aspirates for 10 min at 300× *g*, the cell pellet was resuspended in a hyaluronidase solution (25 IU; SynVitro^®^ Hyadase, CooperSurgical, Trumbull, CT, USA) and kept on a shaker for 3 min. Afterwards, 10 mL of DMEM-F12 (D8062, Sigma-Aldrich, St. Louis, MO, USA) supplemented with 5% (*v*/*v*) human serum (HS; H4522, Sigma-Aldrich, St. Louis, MO, USA) and 1% (*v*/*v*) antibiotic/antimycotic solution (100×; A5955, Sigma-Aldrich, St. Louis, MO, USA) was added, and the samples were centrifuged for 10 min at 300× *g*. Cell pellets were then resuspended and seeded in 6-well cell culture plates (Costar, 3516, Tewksbury, MA, USA) in DMEM-F12 supplemented with 5% (*v*/*v*) human serum and 1% (*v*/*v*) antibiotic/antimycotic solution. Cell cultures were maintained in standard culture conditions (humidified atmosphere, 37◦C, 5% CO_2_), with the cell culture media changed every 3 days. When cell cultures reached confluence of 80–90%, cells were harvested (0.25% trypsin/0.02% EDTA; T4049, Sigma-Aldrich, St. Louis, MO, USA), counted and expanded in 25/75 cm^2^ flasks (EuroClone, Pero, Italy), or used for further experiments.

The isolated and cultured cells were observed under an inverted light microscope (Leica DMI3000 B, Wetzlar, Germany) to monitor their morphological status, and images were taken and analyzed using a digital camera (Leica DMC2900, Wetzlar, Germany) together with the Leica Application Suite (LAS) v.4.5 software.

### 2.4. Flow Cytometry Analysis of GCs

Human granulosa cells at passage 2 were harvested (0.25% trypsin/0.02% EDTA), washed with phosphate-buffered saline (PBS), and fixed for 20 min at room temperature in fresh 4% ice-cold buffered paraformaldehyde. The cells were then washed with PBS and permeabilized with 0.1% Triton X-100 (Merck, Darmstadt, Germany) for 60 min at 4 °C. Following two additional washes, samples containing 1 × 10^5^ cells were incubated for 60 min at 4 °C in the dark with specific anti-human monoclonal anti-LHR (luteinizing hormone receptor; 8G9A2, MA5-31793, Thermo Fisher Scientific, Waltham, MA, USA) and anti-FSHR (follicle-stimulating hormone receptor; 3D5G9, MA5-38525, Thermo Fisher Scientific, Waltham, MA, USA) antibodies. After washing twice in PBS, samples were incubated with secondary goat anti-mouse IgG1 Alexa Fluor^®^ 488 (A-21121, Thermo Fisher Scientific, Waltham, MA, USA) for 30 min at 4 °C. For aromatase cytochrome P450 (CYP19A) detection, cells were directly incubated for 30 min at 4 °C with fluorochrome-conjugated anti-human monoclonal antibody anti-CYP19 (E-9) Alexa Fluor^®^ 647 (sc-374176 AF647; Santa Cruz, Dallas, TX, USA). Non-specific background fluorescence was determined using cells unlabeled with antibodies or appropriate isotype controls. The specific fluorescence labeling was analyzed on a FACS Calibur flow cytometer (BD Biosciences, San Jose, CA, USA) and the results were processed using the Cell Quest Pro v5.1 software (BD Biosciences, San Jose, CA, USA).

### 2.5. Immunofluorescence of GCs

The expression of granulosa cell markers LHR, FSHR, and CYP19A was detected by immunofluorescence (confocal microscopy). Cultured granulosa cells (passage 2, 80% confluency) were trypsinized, counted, resuspended in DMEM-F12 supplemented with 5% (*v*/*v*) human serum and 1% (*v*/*v*) antibiotic/antimycotic solution, and seeded on sterile glass coverslips (3.25 cm^2^) at a concentration of 0.5 × 10^4^ cells/cm^2^. Seventy-two hours later, adherent cells that had reached an optimal density (50–60%) were fixed with 4% paraformaldehyde solution (pH = 7.5) for 10 min and permeabilized with 0.1% Triton X-100 solution (Merck, Darmstadt, Germany) in PBS for 5 min at room temperature. After washing three times with PBS, the cells were labeled for 2 h at room temperature in a water chamber with antibodies against LHR (LHR Monoclonal Antibody (8G9A2), Thermo Fisher Scientific, Waltham, MA, USA), FSHR (Monoclonal Anti-FSHR Antibody (3D5G9), Thermo Fisher Scientific, Waltham, MA, USA), and CYP19A (Anti-CYP19 Antibody (E-9) Alexa Fluor^®^ 647, Santa Cruz, Dallas, TX, USA). This was followed by washing 3 times with PBS and treatment with the fluorochrome-labeled secondary antibody (Goat Anti-Mouse IgG1 Antibody, Alexa Fluor 488, Thermo Fisher Scientific, Waltham, MA, USA), where appropriate. After mounting the coverslips in immersion oil Mowiol (Sigma-Aldrich, St. Louis, MO, USA), the fluorescence reaction was observed on a confocal laser scanning microscope (CLSM, Leica TCS-SPE, Wetzlar, Germany).

### 2.6. MSC Isolation and Culture

Samples of human bone marrow aspirates were obtained from 5 patients (aged 35–72 years) undergoing orthopedic surgical procedures, in accordance with the ethics committee of the Department of Orthopedics and Traumatology, Tzaritza Ioana University Hospital, Sofia, Bulgaria. MSCs from human bone marrow aspirates were isolated and characterized following previously described standard protocols [14]. Cell cultures between the second and fourth passages were used for experimental analysis. MSC samples from each donor were processed and analyzed independently, and no pooling of cells was performed.

### 2.7. Hypoxic and Inflammatory Conditions

In order to establish hypoxic culture conditions, granulosa cells were cultured in a gas mixture consisting of 5% O_2_, 5% CO_2_, and nitrogen at 37 °C within a CO_2_-O_2_-N_2_-regulated incubator. Control cultures were maintained under normal culture conditions (21% O_2_ and 5% CO_2_ at 37 °C).

Inflammatory conditions were induced by supplementing the granulosa cells’ growth medium (DMEM-F12 with 5% (*v*/*v*) human serum and 1% (*v*/*v*) antibiotic/antimycotic solution) with a mixture of human recombinant proinflammatory cytokines: tumor necrosis factor alpha (TNF-α; 50 ng/mL, Sigma-Aldrich, St. Louis, MO, USA), interferon gamma (IFN-γ, Sigma-Aldrich, St. Louis, MO, USA), interleukin-1 beta (IL-1β; 10 ng/mL, Sigma-Aldrich, St. Louis, MO, USA), and interleukin-6 (IL-6; 10 ng/mL, Genaxxon, Ulm, Germany). Control cells were cultured in parallel and without cytokine supplementation.

### 2.8. Cell Culture Experimental Designs

For the direct co-culture with GCs, MSCs (passages 4–5) were harvested and seeded in 6-well plates (Costar, 3516, Tewksbury, MA, USA) and cultivated in DMEM low/10% FBS/antibiotics. Once full confluence was reached, the MSCs were used for subsequent experiments.

For the transwell system, 6-well cell culture inserts (4.5 cm^2^, cellQART, Northeim, Germany) were prepared and seeded with MSCs (passages 4–5), which were cultivated in DMEM low/10% FCS/antibiotics until confluence. The inserts were then used in further assays.

Conditioned medium from MSCs (MSCcm) was collected from cells at passages 4–5 that were seeded in 25 cm^2^ flasks after being expanded until confluence and cultured for 96 h without medium replacement. The collected MSCcm was centrifuged at 300× *g* and either used fresh or stored at −20 °C until further use. MSCcm was used in a 1:1 ratio with fresh medium for the treatment of GCs.

For apoptosis and estradiol secretion assays, granulosa cells were seeded at a density of 5 × 10^4^ cells/cm^2^, whereas, for proliferation assays, GCs were seeded at 2 × 10^4^ cells/cm^2^.

MSC-GC combinations were not fixed across all assays; instead, different donor combinations were used across independent experiments. Within each experiment, the same MSC-GC pairing was maintained for all tested conditions.

### 2.9. Cell Proliferation

For the proliferation assay of GCs, the CellTraceTM Cell Proliferation Kit (Invitrogen, Carlsbad, CA, USA) was used. Briefly, GCs at passage 2 were stained with 1 µmol/L CellTraceTM Far Red dye for 20 min at room temperature in the dark, following the manufacturer’s instructions. Cells were washed and seeded at a concentration of 1 × 10^5^/mL. Control cultures were treated with 10 µg/mL Mitomycin C to inhibit further cell proliferation. Depending on the experimental protocol, GCs were co-cultured directly with MSCs, indirectly using the transwell system, or in the presence of MSC-conditioned media, as previously described. The culture medium was changed after 48 h of incubation. GCs cultured alone were used as a positive control, and Mitomycin C-treated cells were used as a negative control for proliferation. Contact inhibition prevented MSCs from proliferating in co-cultures. On day 4, the cells were harvested and analyzed via the BD FACSCalibur and CellQuest Pro v5.1 software. The mean fluorescence intensity of GCs cultured in optimal conditions was considered as a 100% proliferation rate, according to which the contact and paracrine effects of MSCs in either hypoxic or inflammatory conditions were compared.

### 2.10. Apoptosis Assay

Second-passage granulosa cells were seeded at a concentration of 2.0 × 10^5^ cells per well into 6-well plates (Costar, 3516, Tewksbury, MA, USA) in triplicate and exposed to inflammatory cytokines or hypoxia. Control cultures were maintained in standard growth medium (DMEM-F12 with 5% human serum and antibiotics/antimycotics) and normal culture conditions (21% O_2_; 5% CO_2_; 37 °C). Cells were trypsinized and harvested at 96 h, and the percentage of cells in early (FITC Annexin V-positive and propidium iodide (PI)-negative) and late apoptosis (FITC Annexin V- and PI-positive) was determined using the FITC Annexin V Apoptosis Detection Kit I (BD Biosciences, San Jose, CA, USA), according to the manufacturer’s instructions. Samples were analyzed within one hour on a FACSCalibur flow cytometer (BD Biosciences, San Jose, CA, USA) using the BD CellQuest Pro v5.1 software. All assays were carried out in triplicate and mean values are presented.

### 2.11. Steroid Hormone Assay

Human granulosa cells at the second passage were seeded on 24-well plates (EuroClone, Pero, Italy) at a concentration of 2.0 × 10^4^ cells/well in DMEM-F12/5% human serum/antibiotics and cultured at normal (37 °C, 5% CO_2_) or in hypoxic culture conditions (5% O_2_, 5% CO_2_, and nitrogen, at 37 °C), depending on the experimental design. Cell culture media were collected after 48 h, centrifuged, and stored at −80 °C until the measurement of estradiol production. The concentration of secreted 17β-estradiol in the conditioned media was determined by an electrochemiluminescence immunoassay (ECLIA; Elecsys Estradiol III, ref. 06656021190; Elecsys Progesterone III, ref. 07092539190; Cobas, Roche Diagnostics, Rotkreuz, Switzerland) on the Roche Elecsys 2010 Immunoassay Analyzer (Roche Diagnostics). All assays were performed in triplicate and mean values are presented.

### 2.12. Data Analysis and Statistics

Data and statistical analyses were performed using the SigmaPlot software (v12.5, Systat Software, Inc., San Jose, CA, USA). Data distribution was assessed using the Shapiro–Wilk test. For normally distributed data, statistical significance between groups was evaluated using one-way ANOVA followed by post hoc multiple-comparison analysis with the Holm–Šidák method. In cases where normality was not met, the non-parametric Kruskal–Wallis one-way analysis of variance on ranks was applied, followed by Dunn’s post hoc test with correction for multiple comparisons. Results are presented in the text and figures as mean values ± standard deviation (SD). For all analyses, differences were considered statistically significant at a *p*-value ≤ 0.05.

## 3. Results

### 3.1. Characterization of Human Granulosa Cells

Human granulosa cells were isolated from a total of five clinically healthy donors with a mean age of 33.4 years (age range: 23–40 years), participating in an assisted reproduction program due to male factor infertility or as egg donors. An additional inclusion criterion was the absence of indications of chronic pelvic inflammatory disease.

The optimal in vitro conditions for stimulating and maintaining the proliferative and functional activity of newly isolated granulosa cell lines have been determined. The possibility of their long-term cultivation in quantities and with functional status, allowing the full implementation of the experimental tasks, has been ensured.

Cell isolation from follicular fluid using the procedure described in the Materials and Methods section provided a significant initial quantity of granulosa cells. Small cell aggregates and scattered single adherent cells were observed on the bottom of the culture vessel 24 h after their initial seeding (Figure 2A). The isolated adherent cells varied both in size and shape, exhibiting an elongated or irregular morphology, and had a relatively large nucleus and granulated cytoplasm (Figure 2B). Any erythrocytes that might have contaminated the primary cell culture were completely removed during medium changes and culture harvesting. The granulosa cell samples from each donor were processed and analyzed independently, and no samples were pooled.

The isolated ovarian cells were characterized as GCs through their marker expression profiles and steroid hormone production. Flow-cytometric analysis showed that the vast majority of the cells at passage 2 were positive for FSHR (75.5% ± 4.2), LHR (98.7% ± 0.96), and aromatase cytochrome P450 (CYP450) (94.5% ± 4.4; Figure 2C). The surface and intracellular distributions of CYP450, FSHR, and LHR in donor granulosa cells were also visualized by confocal microscopy. The nuclei of the cells were stained with the fluorescent dye Hoechst (Figure 2D). Furthermore, we found that the cell culture-conditioned media from all tested cultures contained estradiol, with mean concentration values of 314 ± 107.8 pmol/L.

### 3.2. Effects of Mesenchymal Stem Cells on Granulosa Cell Steroidogenesis Under Inflammatory and Hypoxic Conditions

#### 3.2.1. Direct Co-Culture

In the presence of pro-inflammatory cytokines, estradiol production drops substantially compared to the control GCs, confirming that inflammatory conditions impair the steroidogenic activity of granulosa cells (32.29 ± 29.3% vs. 100%; *p* < 0.05). Estradiol levels under normal conditions are set at a baseline of 100%. The presence of MSCs in the culture under inflammatory conditions has no significant effect, as the estradiol levels remain comparable (39.94 ± 6.6% vs. 32.29 ± 29.3%, *p* = ns) (Figure 3A). Under normal culture conditions, MSCs significantly stimulate GC estradiol secretion compared to control levels (205.44 ± 20% vs. 100%, *p* < 0.05). In contrast, hypoxia alone does not significantly suppress estradiol secretion compared to control GCs (87.75 ± 11.6% vs. 100%; *p* = ns), indicating that it may not negatively affect steroidogenesis. However, under hypoxic in vitro conditions, the presence of MSCs markedly increases estradiol secretion compared to hypoxia alone (154.22 ± 11.8% vs. 87.75 ± 11.6%, *p* < 0.05) (Figure 3B).

#### 3.2.2. Indirect Co-Culture

Indirect MSC co-culture in the presence of cytokines (GCs + MSCs + cytokines) increases estradiol production compared to inflammation alone (104.98 ± 8.9% vs. 32.29 ± 29.3%; *p* < 0.05), reaching values closer to those of untreated GCs. MSCs alone (GCs + MSCs) maintain secretion at levels comparable to controls (72.52 ± 15.4%). This suggests that, in an indirect system, MSC-derived soluble factors could support steroidogenesis, even in an inflammatory environment (Figure 3C).

MSC indirect co-culture under hypoxia (GCs + MSCs + hypoxia) modestly improves estradiol levels compared to hypoxia alone, although the differences are not statistically significant (94.62 ± 25.2% vs. 87.75 ± 11.6%; *p* = ns). The trend suggests that MSCs may provide limited support for steroidogenesis in a hypoxic microenvironment when indirect signaling between the two types of cells dominates (Figure 3D).

#### 3.2.3. MSC-Conditioned Medium

When GCs are cultured in the presence of an MSC-derived conditioned medium (MSCcm), estradiol secretion is only slightly elevated compared to that of GCs, showing that, under optimal culture conditions, MSCcm does not have a prominent stimulatory effect on GC steroidogenesis (110.59 ± 4.4% vs. 100%, *p* = ns). When both pro-inflammatory cytokines and MSCcm are added (GCs + MSCcm + cytokines), estradiol secretion is increased compared to GCs treated only with cytokines (73.30 ± 2.3% vs. 32.29 ± 29.3%; *p* = ns), although not to the level of GCs + MSCcm (Figure 3E).

The culture of GCs with MSC-conditioned medium under hypoxia leads to only a slight increase in estradiol secretion, indicating that MSCcm may have a minimal positive effect on GC steroidogenesis in low-oxygen conditions (100.58 ± 4.2% vs. 100%, *p* = ns) (Figure 3F). However, this difference is not statistically significant and therefore represents only a possible trend rather than a confirmed effect.

### 3.3. Effects of Mesenchymal Stem Cells on Granulosa Cell Proliferation Under Inflammatory and Hypoxic Conditions

#### 3.3.1. Direct Co-Culture

GCs under normal in vitro conditions show baseline proliferation (control), which is set at 100%. Proliferation is slightly reduced by cytokine treatment (GCs + cytokines; 80.95 ± 25.6%, *p* = ns). Co-culture with MSCs alone (GCs + MSCs) does not affect proliferation relative to controls (83.5 ± 17.576% vs. 100%, *p* = ns). However, GCs + MSCs + cytokines show the lowest proliferation of all groups (46.91 ± 31.62%)—lower than that of cytokines alone—indicating that MSCs do not rescue—or, in this direct co-culture setting, further impair—granulosa cells’ proliferative capacity in the inflammatory milieu (Figure 4A).

Hypoxia tends to increase proliferation versus controls (112.82 ± 9.8% vs. 100%, *p* = ns). MSC co-culture under hypoxia leads to a lower proliferation rate than under hypoxia alone (77.68 ± 16.72% vs. 112.82 ± 9.8%, *p* < 0.05) (Figure 4B).

#### 3.3.2. Indirect Co-Culture

Indirect MSC co-culture in the presence of cytokines (GCs + MSCs + cytokines) significantly reduces the proliferation rate compared to the controls (53.24 ± 4.3% vs. 100%; *p* < 0.05) and compared to inflammation alone, even though the latter was not statistically significant (53.24 ± 4.3% vs. 80.95 ± 25.6%; *p* = ns). This indicates that MSCs lack the capacity to exert a protective paracrine effect on GCs’ proliferative capacity under inflammatory conditions (Figure 4C).

During hypoxia, proliferation is slightly increased compared to controls (112.83 ± 9.8% vs. 100%; *p* = ns), suggesting that hypoxia may initially stimulate cell growth, although the increase is not statistically significant. The indirect co-culture of MSCs with GCs in a hypoxic environment does not lead to a marked difference in the proliferation of GCs compared to hypoxia alone (109.18 ± 5.7% vs. 112.83 ± 9.8%; *p* = ns) (Figure 4D).

#### 3.3.3. MSC-Conditioned Medium

MSCcm alone decreases GC proliferation, although the difference from the controls is not statistically significant (59.72 ± 3.95% vs. 100%; *p* = ns). In the presence of inflammation, MSCcm does not appear to affect GCs’ proliferation capacity, which remains very similar to that under the effects of MSCcm alone (80.91 ± 1.70% vs. 80.95 ± 25.6%; *p* = ns) (Figure 4E).

Under optimal conditions, MSCcm leads to reduced proliferation compared to the controls (59.72 ± 3.95% vs. 100%; *p* = ns). In a low-oxygen experimental setup, MSCcm inhibits the proliferation of GCs compared to GCs in hypoxia (77.64 ± 19.1% vs. 112.82 ± 9.8%; *p* < 0.05) (Figure 4F).

### 3.4. Effects of Mesenchymal Stem Cells on Granulosa Cell Apoptosis Under Inflammatory and Hypoxic Conditions

#### 3.4.1. Direct Co-Culture

Compared to normal culture conditions, cytokine exposure tends to increase both early and late apoptotic fractions (2.3 ± 2.0% vs. 24.12 ± 18.0% and 6.45 ± 0.2% vs. 11.79 ± 1.1%, respectively; *p* = ns), although these differences do not reach statistical significance and should be interpreted as a trend. MSC co-culture in the presence of cytokines (GCs + MSCs + cytokines) does not reduce either early or late apoptosis; instead, the apoptotic rates are close to or higher than those in GCs + cytokines (22.55 ± 16.0% vs. 24.12 ± 18.0% and 21.54 ± 17.0% vs. 11.79 ± 1.1%; *p* = ns), again without statistical significance, indicating only a tendency toward increased apoptotic rates. This pattern is consistent with MSCs having a negative additive impact on GC survival and proliferation during inflammatory stress (Figure 5A).

Hypoxia elevates the late apoptotic fraction of GCs (12.6 ± 4.4% vs. 6.45 ± 0.2; *p* = ns). There is a clear tendency of MSC co-culture under hypoxia to reduce late apoptosis compared to hypoxia alone (7.74 ± 4.91% vs. 12.6 ± 4.4%; *p* = ns), indicating a possible cytoprotective effect of MSCs in a low-oxygen environment (Figure 5B).

#### 3.4.2. Indirect Co-Culture

Late apoptosis rises further in the indirect co-culture of MSCs with GCs compared to control cells, suggesting that MSCs may not prevent—and may even contribute to—progression into late apoptosis when inflammation persists (24.08 ± 2.6% vs. 6.45 ± 0.2%; *p* < 0.05) (Figure 5C).

MSC co-culture under hypoxia significantly increases late apoptotic fractions compared to control cells (20.72 ± 2.2% vs. 6.45 ± 0.2%; *p* < 0.05), possibly indicating an apoptosis-promoting effect of MSC-derived soluble factors. Early apoptosis was generally not affected by hypoxia or the presence of MSCs in the cell inserts (Figure 5D).

#### 3.4.3. MSC-Conditioned Medium

The addition of MSCcm does not affect early (1.88 ± 0.25%) or late (7.2 ± 0.65%) apoptosis relative to the controls (*p* = ns). However, in the presence of both cytokines and MSCcm (GCs + MSCcm + cytokines), the late apoptotic levels increase to a percentage that remains comparable to that in GCs + cytokines (17.98 ± 0.18% vs. 11.79 ± 1.1%; *p* = ns) but still significantly higher than in the controls (*p* < 0.05) (Figure 5E).

In a low-oxygen environment, MSCcm tends to reduce late apoptosis compared to GCs + hypoxia, but not to a statistically significant level (7.37 ± 2.6% vs. 12.16 ± 4.4%; *p* = ns) (Figure 5F).

## 4. Discussion

Premature ovarian insufficiency remains a serious clinical challenge as there are currently no treatment options available for complete recovery or a partial improvement in ovarian function. Modern medicine most often offers hormone replacement therapy, which reduces the objective symptoms in order to largely ensure a normal and comfortable life for affected women, as well as mitigating the risk of developing complications from ovarian dysfunction, such as osteoporosis and cardiovascular disease [15,16]. For women who wish to become pregnant, the only possible—but, at the same time, a radical—solution to overcome infertility is egg donation. However, for many patients, for various reasons (cultural, social, religious, psychological), this option remains unacceptable. Under these circumstances, it is only natural to look for new solutions to restore normal ovarian function, which would not be achievable without further revealing the specific cellular mechanisms underlying the already observed positive effect of MSCs on the function of granulosa cells. It is important to note that it is precisely under conditions of hypoxia and inflammation that mesenchymal stem cells are activated, adopt an immunosuppressive phenotype, and alter their cytokine secretion [17,18]. In addition, the priming of MSCs through pro-inflammatory cytokines or hypoxia has been used to enhance their therapeutic potential [19,20].

It is well known that systemic and local inflammation, as well as hypoxia, can be extremely damaging to cells, tissues, and the whole organism in general [21]. This especially applies to the ovaries, which are among the organs that are most vulnerable to these conditions. Inflammation in the ovaries is part of normal processes like folliculogenesis and ovulation, facilitating cell proliferation, tissue remodeling, and follicle wall rupture. However, chronic and excessive inflammation can have negative impacts on ovarian health and lead to low oocyte quality, follicular atresia, oxidative stress, POI, and infertility [22]. Inflammatory cytokines can accumulate in the follicular fluid, leading to the activation of the NF-κB pathway and inflammasomes in granulosa cells, causing ovarian dysfunction [23]. Some of the most common triggers of POI are autoimmune disorders, which lead to chronic inflammation that can damage the ovaries [24]. Several studies have investigated MSC-based therapies for POI in animal models of autoimmune diseases, such as systemic lupus erythematosus (SLE) [25,26].

Hypoxia in the ovaries mainly arises from aging-related defects, a disrupted oxygen supply due to surgery, environmental factors, and chronic stress [27]. Although hypoxia-inducible factor 1 alpha (HIF-1α) has an important role in ovarian physiology by regulating the expression of vascular endothelial growth factor (VEGF) and ovarian blood flow [28], chronic and acute hypoxia can lead to follicular atresia and premature ovarian insufficiency by negatively affecting the granulosa cells’ microenvironment and hormonal balance [29].

Most studies working with MSC–granulosa cell interactions and POI have used immortalized human granulosa lines, like HGrC1, which is non-cancerous and has the characteristics of granulosa cells in the early stages of follicular development [30,31]. Most often, ovarian insufficiency is chemically induced in model animals with cyclophosphamide [32,33]. In contrast, we worked with human granulosa cells isolated from preovulatory follicles following a follicular puncture procedure as part of in vitro treatment due to male factor infertility or egg donation. Although primary granulosa cells obtained from IVF patients are more challenging to isolate and cultivate and have limited passage potential in cell cultures, we understand that they retain more in vivo-like functional characteristics in the presence of MSCs and their secreted factors. The isolated cells expressed all three studied phenotypic markers (CYP19A, FSHR, and LHR) and secreted estradiol, which defined them as human granulosa cells.

The objective of this work was to study the interactions between MSCs and granulosa cells under inflammatory and hypoxic conditions and whether MSCs can mitigate the harmful effects of these conditions. In order to achieve this, we set up three experimental models: (1) the direct co-culture of MSCs and GCs, (2) indirect co-culture using transwell inserts, and (3) the cultivation of GCs in the presence of MSC-conditioned media (MSCcm). Although different, these complementary setups allowed us to distinguish between the effects mediated by direct cell-to-cell contact, where cells can continuously communicate via intercellular contacts and soluble factors; indirect contact, where cells can adapt their responses but only through soluble factors; and paracrine signaling in the absence of live cells, where the conditioned medium has prefixed content and concentrations. Recent studies have mostly investigated the effects of MSC-conditioned media or MSC-derived exosomes (MSCexo) as a cell-free treatment alternative with lower risks of complications [34,35,36]. The main disadvantage of this approach remains the exclusion of the possible dynamic intercellular communication (directly or through secreted mediators) that can adapt itself to the microenvironment and can change cellular signals accordingly.

In our study, estradiol secretion appeared as the most consistent functional characteristic affected by MSCs. Inflammation strongly suppressed the estradiol secretion of GCs, indicating the strong inhibitory effect of a pro-inflammatory environment on the steroidogenesis of GCs. However, when GCs were cultured with MSCs (either directly or indirectly) or with MSCcm, estradiol secretion was increased, approaching control levels or even higher, with the strongest effect in the direct co-culture experimental setup [31,37]. Even so, under inflammatory conditions, MSCs in direct co-culture with GCs led to a significant decrease in estradiol production, even falling below control levels. This decline may reflect a contact-dependent shift in MSCs toward an anti-inflammatory phenotype [38]. Interestingly, in hypoxic conditions, estradiol secretion was only slightly reduced compared to the controls, consistent with the possible physiological adaption of GCs to the low oxygen levels needed for normal follicle maturation [39]. MSCs had a positive effect on GC estradiol production in hypoxic conditions when MSCs and GCs were co-cultured directly. It should be emphasized that the hypoxic environment applied here was relative to typical in vitro cultures, where the oxygen concentrations are substantially higher than in vivo. Thus, our hypoxic setting reflected decreased oxygen availability compared to standard culture conditions, rather than a deviation from physiological ovarian oxygen levels.

The effects of MSCs on GC proliferation were more variable, depending on the experimental approach and the culture conditions. Direct co-culture with MSCs did not have an effect on GC proliferation, suggesting that, in the absence of priming factors, MSCs neither enhance nor suppress GC proliferation. Exposure to pro-inflammatory cytokines insignificantly decreased the GC proliferation rate. Direct or indirect co-culture with MSCs did not lead to an increase in proliferation and, in the presence of inflammation, even further suppressed it. This result suggests that, when MSCs are primed with pro-inflammatory cytokines, they may adopt an immunomodulatory phenotype and suppress GC proliferation. In contrast, when MSCcm was used under optimal conditions, granulosa cell proliferation showed a non-significant tendency toward inhibition [12]. However, these effects did not reach statistical significance and therefore represent only potential trends rather than confirmed biological responses.

In contrast, hypoxia tended to stimulate granulosa cell proliferation, possibly reflecting a natural GC response to reduced oxygen availability, which is typical of the follicular microenvironment. Interestingly, MSC co-culture under hypoxic conditions led to a decrease in proliferation relative to hypoxia alone, implying that MSCs may alter their paracrine signaling and adapt a suppressive phenotype [40].

The results relating to the effects of MSCs/MSCcm on GC apoptosis under inflammation and hypoxia were the most variable among the experimental models used. In general, when granulosa cells were exposed to pro-inflammatory cytokines, there was a clear tendency towards increased early and late apoptosis. This finding aligns with the well-established negative effects of inflammation on cells, leading to oxidative stress and caspase activation [41]. Neither direct nor indirect co-cultures with MSCs significantly reduced GC apoptosis, and, in some cases, they even tended to increase it, suggesting that MSCs act as sensors for inflammation, adopt an immunomodulatory phenotype, and may even stimulate cellular apoptosis [42]. Paracrine factors secreted by MSCs were also insufficient to significantly protect GCs from apoptosis. Under hypoxia, direct co-culture and MSCcm slightly reduced late apoptosis, but indirect co-culture increased it, which suggests that the anti-apoptotic effect of MSCs is complex and depends on the type of MSC-GC interaction.

## 5. Conclusions

Numerous studies have shown that MSCs and their secretome can restore ovarian reserves. Although further research is needed to standardize MSC production and assess their long-term safety and efficacy, MSCs are undoubtedly an innovative approach, providing an individualized and minimally invasive treatment for reproductive disorders. Regarding the context of the data obtained, it is important to consider the complex and multifaceted interactions between MSCs and GCs, and we can conclude that, depending on the conditions of the environment and the type of cell signal, MSCs can have diverse effects. Future research should focus on the comparative analysis of different MSC treatment approaches, which could help scientists and clinicians to better understand MSC transformations under inflammation or hypoxia and move closer to developing more effective therapies for patients with ovarian dysfunction.

## Figures and Tables

**Figure 1 biomedicines-14-00027-f001:**
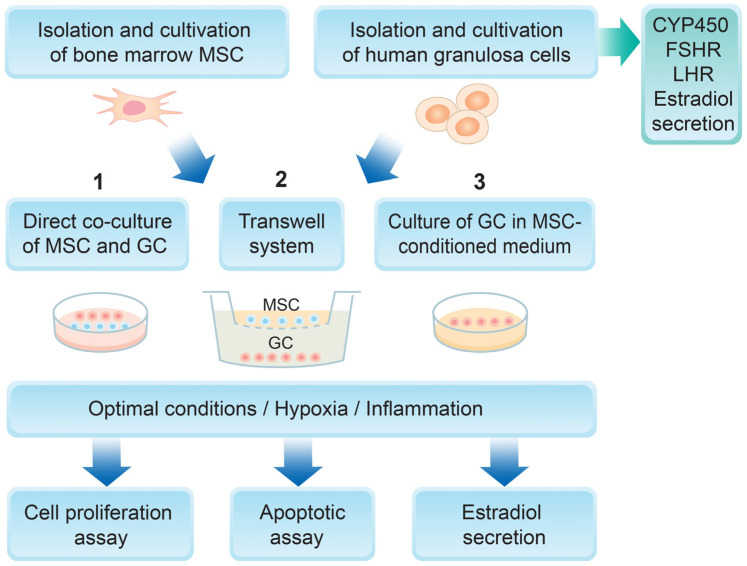
Experimental design of the study.

**Figure 2 biomedicines-14-00027-f002:**
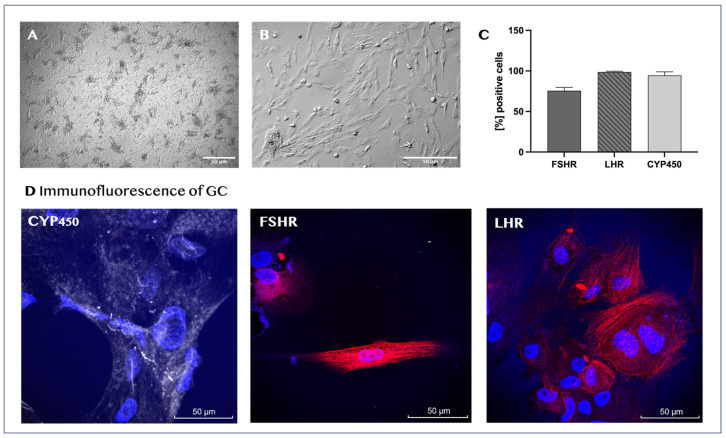
Characterization of human granulosa cells. Representative light microscopy images of the morphological characteristics of human granulosa cells after 24 h of initial culture (magnification, ×60) (**A**) and at passage 4 (magnification, ×160) (**B**). Immunophenotypic profile of newly isolated human granulosa cells. The graph represents the mean values ± SD (*n* = 5) of the percentages for each marker (**C**). Immunofluorescence of isolated granulosa cells (**D**) for CYP19A (gray staining), FSHR (red staining), and LHR (red staining). Cell nuclei are stained in blue.

**Figure 3 biomedicines-14-00027-f003:**
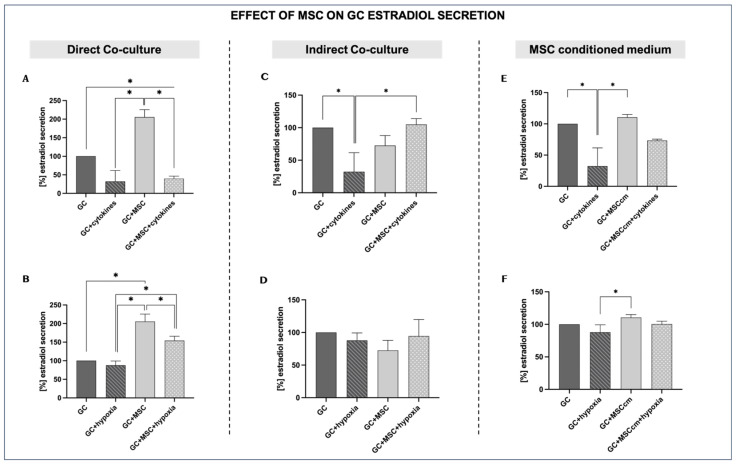
Effects of MSCs on granulosa cell estradiol secretion under inflammation and hypoxia in different culture conditions. For estradiol secretion, GC control results under normal in vitro conditions were set at a baseline of 100%. (**A**,**B**) Direct co-culture of GCs with MSCs under inflammatory (**A**) and hypoxic conditions (**B**). (**C**,**D**) Indirect co-culture of GCs and MSCs under inflammatory (**C**) and hypoxic conditions (**D**). (**E**,**F**) MSC-conditioned medium (MSCcm) treatment of GCs under inflammatory (**E**) and hypoxic conditions (**F**). Data are presented as mean ± SD (*n* = 4). * indicates significant comparisons, *p* < 0.05.

**Figure 4 biomedicines-14-00027-f004:**
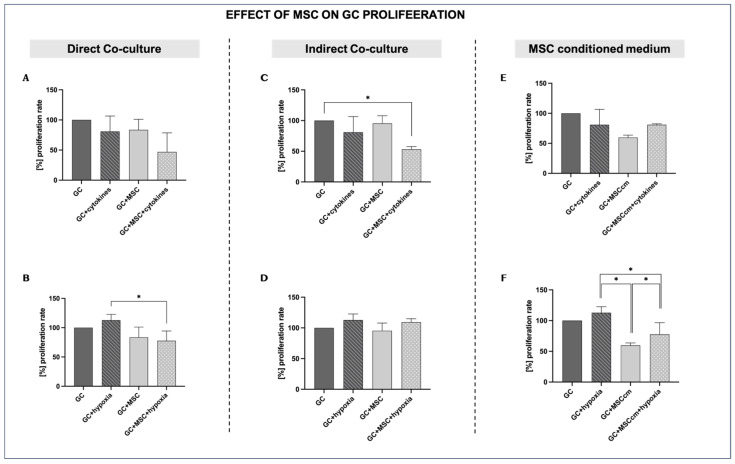
Effects of MSCs on granulosa cell proliferation under inflammation and hypoxia in different culture conditions. For the proliferation rate, GC control results under normal in vitro conditions were set at a baseline of 100%. (**A**,**B**) Direct co-culture of GCs with MSCs under inflammatory (**A**) and hypoxic conditions (**B**). (**C**,**D**) Indirect co-culture of GCs and MSCs under inflammatory (**C**) and hypoxic conditions (**D**). (**E**,**F**) MSC-conditioned medium (MSCcm) treatment of GCs under inflammatory (**E**) and hypoxic conditions (**F)**. Data are presented as mean ± SD (*n* = 4). * indicates significant comparisons, *p* < 0.05.

**Figure 5 biomedicines-14-00027-f005:**
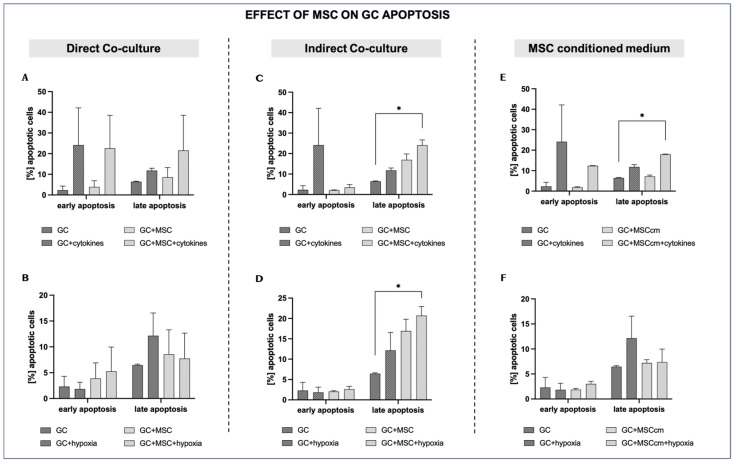
Effects of MSCs on granulosa cell apoptosis under inflammation and hypoxia in different culture conditions. (**A**,**B**) Direct co-culture of GCs with MSCs under inflammatory (**A**) and hypoxic conditions (**B**). (**C**,**D**) Indirect co-culture of GCs and MSCs under inflammatory (**C**) and hypoxic conditions (**D**). (**E**,**F**) MSC-conditioned medium (MSCcm) treatment of GCs under inflammatory (**E**) and hypoxic conditions (**F**). Data are presented as mean ± SD (*n* = 4). * indicates significant comparisons where *p* < 0.05.

## Data Availability

All data of this study are included within the article.

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
