# Peer review of "Mesenchymal Stem Cells Modulate Granulosa Cell Function Under Inflammatory and Hypoxic Conditions"

_biomedicines, 2025, doi:10.3390/biomedicines14010027_

Round 1

Reviewer 1 Report

Comments and Suggestions for Authors   This study addresses a highly relevant issue in reproductive medicine: identifying the mechanisms of therapeutic action of mesenchymal stem cells (MSCs) in ovarian pathologies associated with hypoxia and inflammation. This work is conducted at the intersection of fundamental cell biology and applied regenerative medicine. The authors have successfully identified a research gap: despite encouraging in vivo data, the specific paracrine and cell-mediated mechanisms of MSC influence on granulosa cells (GCs) in various pathological contexts remain insufficiently studied. This topic has high scientific and potential clinical significance. The main strength of the study is the use of three parallel experimental approaches (direct and indirect co-cultivation, as well as culturing with conditioned medium). This design allows us to differentiate the effects of direct cell-cell contacts, paracrine factors, and MSC-expressed vesicles or soluble molecules, significantly deepening our understanding of the observed effects. Moreover, the study is based on modeling the conditions (hypoxia and inflammation) considered key in the pathogenesis of premature ovarian failure, enhancing the value of the data obtained and their potential translational applicability in regenerative medicine. The article is written in good scientific language; all major sections are logical, well-structured, and meet scientific publication standards. The introduction clearly defines the current problem and the specific research gap. The methods are described with a high degree of transparency, allowing for an assessment of the validity of the design and ensuring the potential reproducibility of the study, with the exception of a point regarding statistical analysis that will be addressed below. The results are presented consistently and objectively, and the graphical material is generally well-designed and clearly presented. The reference list includes 40 sources, 22 of which were published in the last five years. Overall, this article is suitable for publication after minor errors are corrected.

Comments:

  • On lines 75-76, the authors write "ability of MSCs to differentiate into various mesodermal and non-mesodermal cell types." It would be better to clarify under what conditions MSCs are capable of differentiating in a non-mesodermal direction.
  • Some parts of Figure 2 lack a scale bar.
  • The lack of mention of correction for multiple comparisons in the "Materials and Methods" section is a significant methodological oversight. The authors should reanalyze the data using the appropriate correction and clearly state the method used in the statistical section. If key findings remain significant after correction, this will only enhance the credibility of the results. If some effects cease to be significant, this should be clearly reflected in the article. This is also a valuable scientific result that prevents misleading readers.

Author Response

Reviewer 1:

  1. Summary

Thank you very much for taking the time to review this manuscript. We are truly grateful to your critical comments and thoughtful suggestions. Based on these comments and suggestions, we have made careful modifications in the original manuscript.

Please find enclosed the revised manuscript, changes made in the manuscript are highlighted. Below you will find our point-by-point responses to your comments.

  1. Point-by-point response to Comments and Suggestions for Authors

  • On lines 75-76, the authors write "ability of MSCs to differentiate into various mesodermal and non-mesodermal cell types." It would be better to clarify under what conditions MSCs are capable of differentiating in a non-mesodermal direction.

Response 1:

We thank the reviewer for this helpful comment. The text in lines 75–76 has been revised to clarify that differentiation of MSC into non-mesodermal lineages occurs under specific inductive conditions, such as exposure to defined growth factors and specialized culture environments. This clarification has been added to the manuscript and is supported by the inclusion of additional references (Refs. 8 and 9).

  • Some parts of Figure 2 lack a scale bar.

Response 2: We have accordingly added scale bars to all light microscopy images and made the scale bars on the immunofluorescence images more visible.

  • The lack of mention of correction for multiple comparisons in the "Materials and Methods" section is a significant methodological oversight. The authors should reanalyze the data using the appropriate correction and clearly state the method used in the statistical section. If key findings remain significant after correction, this will only enhance the credibility of the results. If some effects cease to be significant, this should be clearly reflected in the article. This is also a valuable scientific result that prevents misleading readers.

Response 3: We thank the reviewer for this important comment. Correction for multiple comparisons was applied during the statistical analysis; however, this was insufficiently described in the original version of the manuscript. The “Materials and Methods” section, subsection 2.12. Data Analysis and Statistics has now been revised to clearly state the post hoc tests and correction methods used for multiple comparisons.

Reviewer 2 Report

Comments and Suggestions for Authors

In this study, the authors studied the interaction of granulosa cells and MSCs under inflammatory and hypoxic conditions using various types of in vitro studies. Overall their work was supported by their findings. The manuscript is well-written.

Minor comments:

  • Since two types of cells were used, were the combination of cells (MSC to GC samples) the same for each of the assays conducted, or randomly compared amongst different subjects?
  • Method section, cell culture experimental designs: include more details, for the direct or indirect coculture - the ratio of each cell type; MSC: GC. For the collection of MSCcm, how many cells were seeded? what type of culture plate used? was fresh medium used for the collection of the MSCcm? When using the MSCcm for treatment, was 100% of the CM used, no further dilution?
  • Include scale bars for images in Fig 2 and the sample size.

Author Response

Reviewer 2:

  1. Summary

Thank you for taking the time to review this manuscript. We sincerely appreciate your insightful recommendations and critical remarks. We have made changes to the original manuscript in response to these comments and recommendations. Please find the revised manuscript enclosed. The changes have been highlighted. Our detailed answers to your remarks are provided below.

  1. Point-by-point response to Comments and Suggestions for Authors

  • Since two types of cells were used, were the combination of cells (MSC to GC samples) the same for each of the assays conducted, or randomly compared amongst different subjects?

Response 1: Mesenchymal stem cells (MSC) and granulosa cells (GC) were obtained from independent donors (five donors each). MSC–GC combinations were not fixed across all assays; instead, different donor combinations were used across independent experiments. Within each experiment, the same MSC–GC pairing was maintained for all tested conditions. This clarification was added in “Materials and Methods” section, subsection 2.8. Cell Culture Experimental Designs.

  • Method section, cell culture experimental designs: include more details, for the direct or indirect coculture - the ratio of each cell type; MSC: GC. For the collection of MSCcm, how many cells were seeded? what type of culture plate used? was fresh medium used for the collection of the MSCcm? When using the MSCcm for treatment, was 100% of the CM used, no further dilution?

Response 2: We thank the reviewer for this valuable suggestion. The “Materials and Methods” section, subsection 2.8. Cell Culture Experimental Designs has been revised to provide additional details regarding the experimental cell culture designs. Briefly, MSC were cultured until reaching confluence prior to use in all experimental setups. For apoptosis and estradiol secretion assays, granulosa cells (GC) were seeded at a density of 5×10⁴ cells/cm², whereas for proliferation assays, GC were seeded at 2×10⁴ cells/cm².

For the collection of MSC-conditioned medium (MSCcm), MSC were seeded in T25 culture flasks and expanded until confluence. At that point, the culture medium was replaced with fresh medium, which was collected after 96 hours. The harvested MSCcm was subsequently used for GC treatment in a 1:1 ratio with fresh medium.

These methodological details, including culture vessels, seeding densities, medium replacement, and conditioned medium usage, have now been clarified in the revised manuscript.

  • Include scale bars for images in Fig 2 and the sample size.

Response 3: We have accordingly added scale bars to all light microscopy images and made the ones on the immunofluorescence images more visible.